# A Validation Study to Confirm the Accuracy of Wearable Devices Based on Health Data Analysis

**Nikola Hrabovska [1,2,\*]**, **Erik Kajati [1]** and **Iveta Zolotova [1,\*]**

1 Department of Cybernetics and Artificial Intelligence, Faculty of EE & Informatics,
Technical University of Kosice, 042 00 Kosice, Slovakia; erik.kajati@tuke.sk
2 Institute of Informatics, Slovak Academy of Sciences, 831 01 Bratislava, Slovakia
\* Correspondence: nikola.hrabovska@tuke.sk (N.H.); iveta.zolotova@tuke.sk (I.Z.)

**Abstract:** This research article presents an analysis of health data collected from wearable devices, aiming to uncover the practical applications and implications of such analyses in personalized healthcare. The study explores insights derived from heart rate, sleep patterns, and specific workouts. The findings demonstrate potential applications in personalized health monitoring, fitness optimization, and sleep quality assessment. The analysis focused on the heart rate, sleep patterns, and specific workouts of the respondents. Results indicated that heart rate values during functional strength training fell within the target zone, with variations observed between different types of workouts. Sleep patterns were found to be individualized, with variations in sleep interruptions among respondents. The study also highlighted the impact of individual factors, such as demographics and manually defined information, on workout outcomes. The study acknowledges the challenges posed by the emerging nature of wearable devices and technological constraints. However, it emphasizes the significance of the research, highlighting variations in workout intensities based on heart rate data and the individualized nature of sleep patterns and disruptions. Perhaps the future cognitive healthcare platform may harness these insights to empower individuals in monitoring their health and receiving personalized recommendations for improved well-being. This research opens up new horizons in personalized healthcare, transforming how we approach health monitoring and management.

**Keywords:** wearable devices; health data analytics; data mining; Internet of Things; wireless transmission; physical activity

## 1. Introduction

The rapid advancements in wearable technology have opened up new avenues for monitoring and analyzing health data. In this era of personalized healthcare, understanding the potential benefits of wearable devices in assessing individual health status becomes increasingly crucial. This research article presents a pioneering analysis of health data collected from wearable devices, aiming to shed light on the practical applications and implications of such analyses. Health could be defined as a combination of complete physical, mental, and social well-being [1]. Today, due to the increase in sluggish living behaviour, a decrease in physical activity leads to health problems. The beneficial effects of regular physical activity on many health outcomes are well known [2]. Research has demonstrated specific benefits such as improvements in physical and physiological health parameters [3,4]. Several lines of evidence have also shown that physical activity can effectively improve mental well-being and reduce the potential for preventing symptoms of mental health disorders such as depression and anxiety [5–8]. We see great potential to focus on health data analytics based on global impacts. We know that various external factors significantly impact human health, and there is a need to explore different cases and collaborate with different sectors for humanity to live sustainably and with positive health [9–11].

Nowadays, wearable devices can collect physical activity data, and coupled with the analytical capabilities provided by artificial intelligence and machine learning, they can potentially raise our level of taking care of ourselves. Wearable devices such as activity trackers and wearable devices can provide unique insights into our health and well-being. Unlike conventional testing in a clinical setting, which may occur a few (or fewer) times a year, wearables offer continuous access to real-time physiological data. This allows deviations from a person's "usual" baselines to be detected: an approach to healthcare that is fundamentally different from current practice, which predominantly compares physiological measurements to population statistics.

Wearable technology differs from mobile devices and is designed to be imperceptible for everyday use [12]. The development of this type of technology is changing how medicine is practiced and how healthcare is delivered. Patients can now measure personal metrics such as heart rate, blood pressure, blood sugar, oxygen levels, body temperature, and other vital signs, improving their lifestyle. This research article represents a pioneering exploration of the practical applications of wearable device data analysis in the realm of health monitoring. By delving into heart rate, sleep patterns, and specific workouts, the study reveals valuable insights that can transform personalized healthcare approaches and lay the groundwork for the development of innovative cognitive healthcare platforms. These advancements have the potential to revolutionize the way we understand and manage our health, placing individuals at the forefront of their well-being journey.

## 2. Related Works

With the development of information technology, smart healthcare is gradually coming to the fore. Smart healthcare uses next-generation information technologies such as the IoT, big data, cloud computing, and AI to comprehensively transform the traditional healthcare system, making healthcare more efficient, convenient, and personalized [13–16]. The Internet of Medical Things (IoMT) emerges as a next-generation bio-analytical tool, which was created by the development of intelligent sensors, smart devices, and advanced lightweight communication protocols, which allow medical devices to be connected to monitor biological signals and diagnose patient ailments without using humans [13,17]. Articles from this area of research can be divided into two groups, with one group containing articles that focus more on the actual functionality of these devices and their accuracy, and the other group focusing more on the literature within healthcare and the linking of AI to healthcare, for further advancements in the field. The most frequent or the most significant number of articles are based on the context of using a given AI method already with specific medical diseases.

Wu et al. [18] present a gadget that is used to see the relationship between activity and energy expenditure, which creates a paradigm for the use of wearable devices for personalized prevention. Altini et al. [19] conclude that resting heart rate and heart rate variability (HRV) also can effectively be used to quantify individual stress responses across a large range of individual characteristics and stressors. A recent paper, where an algorithm is proposed to determine the current health status of a patient, introduces a COVID-19 patient health management platform that uses the IoT and cloud computing technology [20]. Ambulatory monitoring devices, including wearables, smartphones and other ambulatory sensors, enable a new healthcare paradigm by collecting and analyzing long-term data for a reliable diagnosis. In 2016, Wallen et al. [21] compared and examined the accuracy of wearable devices, where correlations between reference methods and devices were moderate to strong for heart rate and weak to strong for energy expenditure. As a result, wearable devices accurately measure heart rate, but the estimates for energy expenditure are pretty poor. This analysis includes the Apple Watch wearable device. These watches have been popular for a long time and are still in high demand, increasing researchers' interest in analysing these wearable devices [22]. However, the paper above does not have data that would be measured during higher-intensity exercises, where the accuracy of these wearables may also be reduced due to higher movement intensity. A major advantage over

conventional testing in a clinical setting, which may be only a few times a year or les. In a clinical setting, the number of times conventional testing can be performed depends on the type of test and the patient's condition. In general, the number of times a test is performed is determined by the healthcare provider based on the patient's needs and the diagnostic value of the test. The healthcare provider will consider factors such as the patient's symptoms, medical history, and current condition, as well as the risks and benefits of the test. Wearable devices offer continuous access to physiological data in real-time. Wearable devices are also beneficial in situations where, for example, a coronavirus pandemic (COVID-19) breaks out and mankind was faced with an as-yet-unknown phenomenon. When we have no historical data and do not know what lies ahead, this is when wearables can help us. When worn correctly, they can contribute to pre-pandemic data while storing new data suitable for research [23–25]. Mishra et al. [26] showed that we could use data from wearable devices for the presymptomatic detection of COVID-19. An analysis of physiological and activity data from 32 users with a positive result for this disease was performed, where it was found that up to 81% had changes in heart rate, number of steps per day or sleep time. Up to 63% of COVID-19 cases could be detected before the onset of symptoms in real time by whether we see an increase in resting heart rate compared to an individual baseline. This finding shows that wearable devices are a valuable tool for the real-time detection of respiratory infections. We can also read in the article that remote voltage measurement is fundamental given the transition of life from offline to online caused by the COVID-19 pandemic. Sun et al. [27] describe remote physiological signal features that provided higher accuracy in stress estimation (78.75%) compared to features based on motion (70.00%) and facial expression (73.75%). In addition, the fusion of behavioral features and remote physiological signals increased the accuracy of stress estimation up to 82.50%. However, based on the literature analysis, we see a significant gap in the study of health data from wearable devices, specifically from a large and underutilized database typically stored on each user's mobile device. We see great potential in investigating health data from wearable devices, where healthcare personnel could have access to health data that would otherwise be inaccessible. The main problem is that we do not provide healthcare professionals with the information that we see on our mobile devices because it is difficult to describe, for example, our lifestyle verbally. In such cases, simple statistical analyses provided by mobile devices in collaboration with wearable devices are very beneficial. This is convenient for both users, who have no role other than utilizing these devices correctly, and for healthcare personnel, who directly receive graphically represented results. Our motivation is to explore the potential of health data from wearable devices for total utilization. Articles devoted to such analyses are not available. Based on previous studies and our experience, we can say that analyzing data from wearable devices has several challenges, including:

- Data Quality: Wearable devices may produce large amounts of noisy or incomplete data, which can affect the accuracy of the analysis.
- Data Integration: Data from wearable devices may need to be integrated with other data sources to provide a complete picture of the user's health status, which can be challenging due to differences in data formats and data standards.
- Data Privacy and Security: Wearable devices may collect sensitive data, such as personal health information, which requires appropriate privacy and security measures to protect against unauthorized access or disclosure.
- Data Interpretation: The interpretation of data from wearable devices can be complex, requiring expertise in data analytics, machine learning, and other advanced techniques to extract meaningful insights.
- User Engagement: Users of wearable devices may lose interest in wearing them over time, resulting in incomplete or inconsistent data.
- Technical Limitations: Wearable devices may have technical limitations that affect data collection, such as limited battery life, signal interference, or device failure.

- Bias: Wearable devices may have inherent biases that affect the accuracy and validity of the data, such as differences in device placement, sensor type, or user demographics.

In this case, this can affect monitoring patterns. It is also important to note that article [26] is investigating a specific disease, and we do not yet know how to distinguish COVID-19 presymptomatic state from other respiratory inflections. Many methods have been developed to predict diseases using IoT, but less often do we see the combination of AI and IoT devices. For example, in [28], Muthu et al. present a machine learning-based neuro-fuzzy algorithms where they train data into AI classification using the deep learning mechanism of Boltzmann's Belief Network. In the proposed methodology, they achieved a 96% prediction rate and 96.33% accuracy.

We see in various examples the importance of incorporating wearable devices in detecting and investigating various diseases or patient health conditions. Therefore, another aspect of the research is the need to evaluate the accuracy of these devices. Wearable devices often use low-cost and easy-to-use photoplethysmography (PPG) technology [29]. It is used to measure vital signs, including heart rate (HR) and pulse rate variability (PRV), which is used as a surrogate for HRV. Blok et al. [30] investigated the accuracy of a wearable device, the Cardiowatch 287, a certified wireless remote monitoring system designed for continuous physiological data collection in the home environment and healthcare settings. They investigated the accuracy of a wearable device for heartbeat detection in cardiac patients and evaluated the efficacy of a signal qualifier in identifying medically valuable signals. In order to compare the accuracy of the PPG sensor for detecting heartbeats within 100 and 50 ms of the electrocardiography (ECG)-detected heartbeats, as well as correlation and Limits of Agreement for heart rate (HR) and RR-intervals, patients from an outpatient cardiology clinic underwent a simultaneous resting ECG and PPG recording, where ECG sensors measure the bio-potential generated by electrical signals that control the expansion and contraction of heart chambers. In 180 patients 7914 ECG-, and 7880 (99%) PPG-heartbeats are recorded. The PPG-accuracy within 100 ms was 94.6% and 89.2% within 50 ms. Correlation was high for HR (R = 0.991 (95% CI 0.988–0.993), $n$ = 180) and RR-intervals (R = 0.891 (95% CI 0.886–0.895), $n$ = 7880). More about the RR interval in peer-reviewed studies [31]. The HR and RR-interval 95% Limits of Agreement (LoA) were 3.89 to 3.77 (mean bias 0.06) and 173 to 171 (mean bias 1) beats per minute, respectively. They show that this wearable device with PPG-technology can accurately identify the HR and RR intervals in the cardiovascular at-risk patient population across various subgroups. Few studies examine the accuracy of wearable devices during any sporting activity. This issue was also encountered by Støve et al. [32], who decided to contribute research where they assessed the accuracy of the Garmin Forerunner 235 wearable device during various exercises of varying intensity compared to the Polar RS400 benchmark. A total of 29 participants participated in the study, where they had their HR measured using the aforementioned wearable device Garmin during rest and three submaximal exercises; bicycling, walking on a treadmill, running and rapid arm movements. The results of the current study demonstrated that the Garmin Forerunner 235 produced accurate readings of heart rate during rest, treadmill running, rapid arm movement, and cycling at 150 W, although results varied depending on the type and intensity of the exercise, and measurement latency may have an impact on how well it works.

Although articles that are more focused on any health disease are more relevant, especially in the healthcare sector, they also require more expertise from the researchers and much more precise outputs, i.e., the more precise the outputs, the more relevant the article itself is. Articles that describe the functionality of wearable devices in healthcare and their accuracy also provide background information for other researchers doing more specialized work to keep up to date with new devices and their updates.

### 3. Methodology

Despite the enormous amount of data we generate daily, only 0.5% of it is analyzed and used for discovery, improvement, and information extraction [33]. Data analysis is the process of collecting, modeling, and analyzing data to extract insights that support decision-making. There are several methods and techniques for conducting the analysis, depending on the industry and the research objective. All those methods are based on two main areas: quantitative and qualitative research. This work involves both quantitative and qualitative research.

#### 3.1. Data Mining

Data mining is one of the data analysis methods we use in this work. Using exploratory statistical evaluation, data mining aims to identify dependencies, relationships, patterns, and trends to generate advanced knowledge. Adopting a data mining mindset is essential for success when analyzing data.

The most commonly used methodology in practice and the most cited in the literature is CRISP-DM (CRoss-Industry Standard Process for Data Mining) [34]. CRISP-DM is a process consisting of 6 distinct phases that are approached sequentially at a nominal level, but the process itself is iterative, meaning that all models and understanding are designed to be improved by subsequent insights gained throughout the process. CRISP-DM is illustrated in Figure 1 below and work as follows:

- Business understanding—the first stage is to understand the project objectives and identify specific requirements truly.
- Data understanding—once the project goals are defined, the process of understanding the data will begin.
- Data preparation—the final dataset is being prepared for modeling at this stage.
- Modelling—we select and use different modeling techniques and their parameters calibrated to optimal values.
- Evaluation—in the evaluation phase, we are already looking at which model best meets the project's requirements and what to do next.
- Deployment—includes deployment plan, final report or project summary and project retrospective.

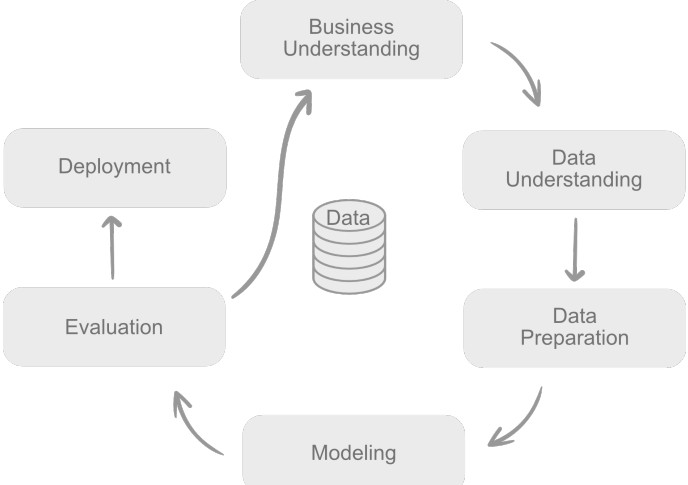

**Figure 1.** CRISP-DM methodology diagram.

#### 3.2. Data Preprocessing

Input data was collected from 5 respondents who use Apple Watch watches. The study aimed to investigate the feasibility and potential benefits of utilizing such devices to monitor and improve personal health. Despite the limited number of respondents, this article showcases several compelling reasons for the smaller sample size.

- Technological constraints and availability: At the time of conducting this study, wearable devices were relatively expensive or not widely distributed among the general population in a relevant country. By wanting to examine data from specific Apple Watch devices, the sample size of the data was greatly affected. In addition, collecting health data from wearable devices is generally challenging because we can only access it directly through the wearers of such devices in most cases. This means we cannot rely on, for example, hospitals within which collaboration could be established.
- Informed consent and privacy concerns: Given the sensitive nature of health data, participants have hesitated to share their personal information, even with guaranteed anonymity and strict data security measures. This apprehension towards data privacy and security concerns has significantly limited the number of respondents willing to participate, leading to a smaller sample size.
- Rigorous data collection and quality assurance: By having a smaller sample size, we were able to invest more time and effort in ensuring the accuracy, reliability, and validity of the data collected. With a limited number of respondents, we could focus on maintaining a higher level of data quality control, thereby increasing the reliability of our findings.
- Deep analysis and individual focus: With a smaller sample size, we conducted a more in-depth analysis of each participant's health data. This approach gave us a comprehensive understanding of individual health patterns, outliers, and potential correlations, providing valuable insights that might have been overlooked in studies with larger sample sizes.

In order to comply with GDPR, the General Data Protection Regulation, we have included, in Table 1, all the information that respondents contributed to this analysis. In order to better identify the respondents during our analysis, each one was given an identification number by which we will distinguish them in the following sections of the paper, where the first number indicates the gender of the respondent (1—female, 2—male), and the second is a randomly generated number. The Apple Watch collects various types of data that can provide insights into health, fitness, and daily activities. In our dataset, the attribute "number of records" represents the following data:

- Activity Data: This category includes step count, distance traveled, active calories burned, and other activity-related metrics. For example, respondent ID-11 has 40% of the records related to activity information.
- Heart Rate Data: Heart rate measurements are often collected continuously or periodically by the Apple Watch. For example, respondent ID-11 has 40% of the records related to heart rate information.
- Workout Data: When you engage in specific workouts or exercises, the Apple Watch can track metrics such as duration, distance, average heart rate, and calories burned for activities such as running, cycling, swimming, and more. For example, respondent ID-11 has 19% of the records related to activity information.
- Sleep Data: With sleep tracking features, the Apple Watch can gather data about sleep patterns, including the duration and quality of sleep, time spent in different sleep stages, and sleep disruptions. For example, for respondent ID-11, from the "number of records" attribute, we have 349 days of sleep measurements.

**Table 1.** Respondents and their data.

| ID | Gender | Age | Number of Records |
|----|--------|-----|-------------------|
| 11 | Woman | 24 | 1,660,041 |
| 12 | Woman | 24 | 975,976 |
| 23 | Man | 20 | 1,187,518 |
| 24 | Man | 25 | 2,379,368 |
| 25 | Man | 31 | 3,679,617 |

We use the R programming language during our work for several reasons:

- Open Source: R is an open-source programming language, which means that it is free to use and can be easily customized and modified to meet specific needs.
- Wide Range of Packages: R has a wide range of packages and libraries that make it easy to perform complex statistical analyses, data visualization, and machine learning tasks.
- Community Support: R has a large and active community of users and developers who contribute to the development of new packages, share best practices, and provide support through online forums and user groups.
- Data Visualization: R has powerful data visualization capabilities that make it easy to create high-quality charts, graphs, and other visualizations that help to communicate complex data insights.
- Reproducibility: R allows for reproducible research, meaning that data analysis workflows can be documented and shared with others, allowing for transparency and collaboration.
- Integration with Other Tools: R can be easily integrated with other tools and technologies, such as databases, web applications, and other programming languages, making it a flexible and versatile tool for data analysis.

Overall, R is a powerful and flexible tool for data analysis that is widely used by researchers, data scientists, and other professionals in a variety of fields, including academia, industry, and government. One of the first challenges is to navigate the huge amount of data available. Initially, we need to download all the necessary libraries based on what we specifically want to do with the acquired data. We use the *tidyverse*, *lubridate*, *scales*, *ggthemes*, and *ggplot2* packages to create elegant visuals that present our exploratory analysis of data from 1 person's Apple Watch wearable device.

For each respondent, we can confirm that the wearable device was used for casual wear and to record daily activity, such as sleep and steps, and occasional sporting activity, such as running or strength training. We focus on examining several factors that have been measured by the wearable device: active energy burned, heart rate, and sleep. Because respondents have different numbers of records, we chose an interval of 1 year, which we subsequently analyzed. We do not use all the attributes, but the key ones are as follows:

- *Record*—Where all the data is stored, whether some basic information about the user, such as weight and height, or information recorded by the watch, such as heart rate or steps.
- *Workout*—Here, we can find all the records of the exercises we have run through our wearable device.
- *Activity Summary*—Daily activity statistics. Apple Watch is also known for providing its users with daily goals through 3 circles: moving, exercising, and standing.
- *Instantaneous Beats Per Minute*—Instantaneous heart rate recordings.

In the data preprocessing, we only made some basic adjustments, such as removing duplicates, empty "white" spaces or adjusting the time zone. However, those adjustments varied according to the analysis given. The time zone only needed to be changed if the particular data was exported in a different time zone than the one we were currently in. In fact, the time zone does not represent the current time zone as we would like it to. The time zone attribute stores the value of when the data was exported, i.e. in which time zone.

## 4. Results

The maximum heart rate (MHR) during exercise varies from person to person and is influenced by various factors such as age, fitness level, and genetics. The commonly used formula to estimate MHR is 220 minus age. For example, if a person is 30 years old, their estimated MHR would be 190 beats per minute ($220 - 30 = 190$) [35].

However, this formula provides only an estimate and may not be accurate for everyone. It is also important to note that exercising at your maximum heart rate may not be safe

for everyone and should be performed under the guidance of a healthcare professional. During exercise, the American Heart Association (AHA) suggests using the talk test as a simple way to monitor exercise intensity. If you can talk comfortably during exercise, you are likely working within the recommended target heart rate zone. If you are too breathless to talk, you may be working too hard and should consider slowing down.

In general, during exercise, it is recommended to maintain a heart rate within a target range of 50–85% of your estimated MHR [36]. This range is considered safe for most people and can help improve cardiovascular fitness and endurance. It is recommended to consult a healthcare professional or a certified personal trainer to determine a safe and effective target heart rate range for exercise.

On average, the AHA recommends target heart rates during exercise as follows in Table 2.

**Table 2.** Heart rate target zone by age (AHA).

| Age | 20 | 30 | 40 | 50 | 60 | 70 |
|---|---|---|---|---|---|---|
| HR (BPM) | 100–170 | 95–162 | 90–153 | 85–145 | 80–136 | 75–128 |

A 2018 review study [8] shows us that people can improve their heart health and reduce resting heart rate through regular exercise. Regular exercise reduces the risk of heart attack, stroke and other health problems. However, researchers also suggest that consistently high levels of exercise—such as marathon running—may harm heart health.

*4.1. Heart Rate*

Engaging in aerobic and endurance exercises also helps to improve fitness, increase muscle tone and improve overall physical and mental well-being. In fact, one 2016 meta-analysis [37] reported that:

**Hypothesis 1.** *Exercise has a large and significant antidepressant effect on people with depression.*

We were in contact with respondent ID-11 throughout the development of this research. As a result, we had the opportunity to present his results to him in person and receive other relevant information to interpret our results better. In Figure 2, we can see the average BPM values broken down by specific dates, where we are most interested in the break in mid-March 2021. We have distorted the data and plotted it from the date 19 November 2020 to 2 June 2021 to make the results more visible. Based on the feedback, we found that the respondent had a medical condition between 19 November 2020 and 2 March 2021 with problems that were the reason for his limited physical activity. Values changed daily, and up to 65% increase in average heart rate compared to the previous value. The respondent did not suffer from depression directly. However, he did have a mentally challenging period in the interval when he had health problems. Based on his information, we can confirm that exercise significantly impacts an individual's mental health. The World Health Organization (WHO) defined health in 1948 as a state of complete physical, mental, and social well-being and not merely the absence of disease or infirmity [38]. Using observational data, we can say that movement alone positively impacts an individual's mental health. This somehow contributes to the confirmation of Hypothesis 1. Heart rate can increase significantly even from simple activity such as walking. In this case, the respondent's inclusion of the Walking Outdoors exercise in his activity resulted in a significant increase in his heart rate compared to the previous period. When you walk, your muscles require more oxygen to work, and your heart has to pump more blood to deliver oxygen to those muscles. This increased demand for oxygen and blood results in an increase in heart rate.

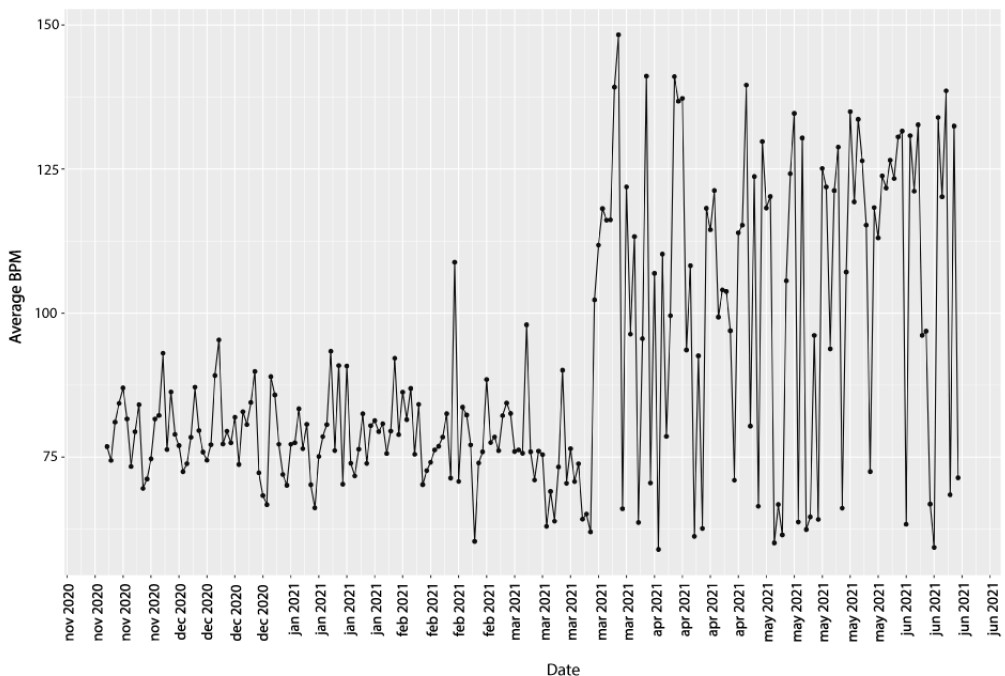

**Figure 2.** Average BPM values and date range.

The heart rate of an individual varies based on various motor activities. At Figure 3, we can see a comparison of 2 physical activities based on their maximum heart rate. The picture shows how these 2 movement activities differ based on the user's heart rate. Swedish researcher Gunnar Borg [39] developed the Borg Rating of Perceived Exertion (RPE), a tool for measuring an individual's effort and exertion, breathlessness and fatigue during physical work. Simply put, it measures how hard the body feels like it is working based on the bodily sensations that the subject feels, such as increased heart rate, respiration or breathing rate, increased sweating, and muscular fatigue. Although subjective, rating exertion on a rating scale of 6 to 20, as seen in Table 3.

**Table 3.** BORG rating of perceived exertion (RPE).

| Score | Level of Exertion |
|---|---|
| 6 | Not exertion at all |
| 7 | |
| 7.5 | Extremely light |
| 8 | |
| 9 | Very light |
| 10 | |
| 11 | Light |
| 12 | |
| 13 | Somewhat hard |
| 14 | |
| 15 | Hard (heavy) |
| 16 | |
| 17 | Very hard |
| 18 | |
| 19 | Extremely hard |
| 20 | Maximal exertion |

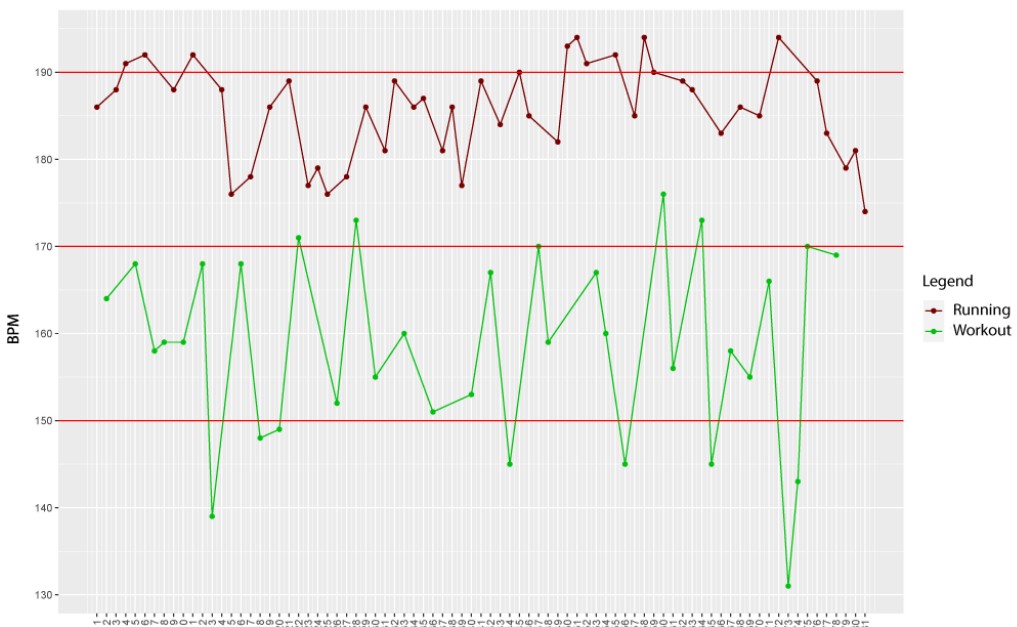

**Figure 3.** Max BPM values during functional strength training (workout) and running.

**Hypothesis 2.** *Can provide a pretty good estimate of the heart rate during physical activity.*

There is a high correlation between perceived exertion rating times 10 and actual heart rate during physical activity. For example, if a person's RPE is 13, then $13 \times 10 = 130$; so the heart rate should be approximately 130 beats per minute. To conduct this study, we recruited a physically active participant with no underlying health conditions. We monitored her heart rate using a wearable device throughout the exercise routine and asked her to rate their perceived exertion level on the BORG RPE scale. After the exercise, we recorded the heart rate data and compared it to the participant's perceived exertion level on the BORG RPE scale. We have shown the results in Table 4.

**Table 4.** Rate of perceived exertion.

| RPE score | 10 | 11 | 13 | 15 | 16 | 17 |
|---|---|---|---|---|---|---|
| **HR (BPM)** | 107 | 112 | 129 | 145 | 154 | 162 |

The results are recorded at the same time. We can see at a glance that the results at higher heart rates are more distant from the results with the participant's level of perceived exertion on the BORG RPE scale. However, these variations are still minor enough that this scale can be investigated to yield new insights into the individual's perceived exertion. We found that the heart rate data and the BORG RPE scale ratings were approximately the same. This indicates that the BORG RPE scale is an effective tool for measuring an individual's perceived exertion level during exercise and approximately reflects their heart rate. In the same way, with wearable devices, users are alerted to calorie burn results, which are also only approximate values. This data should be used with some respect, and therefore it is also recommended to consult a doctor if we want to make any changes to our lifestyle based on this information.

Our findings confirm the Hypothesis 2 that the BORG RPE rating scale provides approximately accurate heart rate values during exercise. This information is crucial for exercise professionals and individuals who monitor their exercise intensity using heart rate monitors, as it can help them make more informed decisions about their workout routine and ensure they are working at the appropriate intensity level. Future studies can build upon our findings and investigate the effectiveness of the BORG RPE scale in

other populations and exercise intensities. Furthermore, the BORG RPE rating scale is a practical tool for measuring exercise intensity, as it does not require any equipment or special training, making it accessible to a wide range of individuals.

Based on these results and interactions with the respondent, we can confirm the accuracy of the wearable device based on the observed heart rate attribute. Based on the medical data analysis regarding heart rate and the processed study regarding the target heart rate zone during exercise, we observed Respondent ID-11 had average heart rate values during Functional Strength Training that fell within the norm of the target zone during exercise. The target heart rate zone during exercise is the range of heart rate that you should aim for during physical activity to achieve the most cardiovascular benefit from your workout. This range is typically defined as a percentage of your maximum heart rate, which is the highest number of times your heart can beat per minute. Consistent moderate or vigorous running gets our heart rate up, thus increasing its longevity and capacity in the long term [40]. Exercising will strengthen your heart and lungs, but most importantly, your resting heart rate will begin to decrease [41]. Next, we presented the heart rate results between Functional Strength Training and Running, where we saw significant differences in maximum heart rate values, which confirms the fact that there is a visible difference in exercise intensity between the two workouts.

*4.2. Sleep*

Wearable devices can collect sleep data objectively and continuously. This means that the data is not subject to the biases and inaccuracies that can occur with self-reported sleep data. Continuous data collection also means capturing important changes in sleep patterns over time, which can provide valuable insights into an individual's overall health and well-being. Wearable devices are non-invasive and easy to use, making them a convenient option for sleep analysis. Sleep data collected from wearable devices can help identify sleep disorders and other health issues at an early stage. For example, changes in heart rate during sleep can be an early indicator of cardiovascular disease, while disruptions in sleep patterns can be a sign of sleep apnea or other sleep disorders.

During the analysis of the health data, we focused on the temporal interface and heart rate. Studies consist of analysing sleep onset, sleep end, sleep duration, optimal sleep duration, and sleep interruption. We look at the average values of heart rate during sleep and the time of recording the minimum heart rate during sleep.

4.2.1. Optimal Sleep Duration

The optimal length of sleep varies depending on factors such as age, lifestyle, and overall health. However, most adults generally need between 7 and 9 h of sleep per night to function at their best during the day [42]. It is important to note that individual sleep needs can vary based on genetics, age, and lifestyle. For example, some people may feel rested and alert after only 6 h of sleep, while others may require 10 h to feel fully rested. Lifestyle factors such as exercise, diet, and stress can also affect sleep needs. It is also important to consider the quality of sleep in addition to the length of sleep. Even if you get the recommended amount of sleep each night, poor sleep quality can lead to daytime fatigue, mood disturbances, and other health issues.

In our case, all respondents fall into one age category where the optimal length of sleep is between 7 and 9 h. In Table 5, we have shown the original state, where we see the number of sleep records from specific respondents. The number of records, in this case, represents the number of nights recorded as sleep. Based on the available sleep data, we selected the 3 respondents with the most considerable amount of data to use for analysis.

**Table 5.** Sleep data.

| ID | 11 | 12 | 23 | 24 | 25 |
|---|---|---|---|---|---|
| **Number of Records** | 349 | 11 | 327 | 0 | 520 |

We applied seven analyses to the three selected respondents, i.e., their data. We first looked at the onset of sleep. When the wearable device begins to store sleep data, the device detects that the respondent has fallen asleep. Based on the feedback from the respondents, we can tell the correctness of the sleep detection by the device.

We can determine whether the respondent gets the optimal amount of sleep based on sleep duration alone. However, we do not know the record incidence at other intervals. In Table 6, we list the time interfaces that contain records from a given respondent. We have shown the percentage of occurrence of records in the given intervals.

**Table 6.** Optimal sleep duration.

| ID | 2–4 h | 4–6 h | 6–7 h | 7–9 h | 9–10 h | 10–12 h |
|----|-------|-------|-------|-------|--------|---------|
| 11 | 2% | 6% | 19% | 60% | 7% | 2% |
| 23 | 8% | 27% | 28% | 37% | 0% | 0% |
| 25 | 0% | 2% | 7% | 86% | 4% | 0.2% |

The best results are seen for respondent ID-25, where most of the values fall into the indicated category of optimal sleep duration. For respondent ID-23, we see 2 time intervals for which the occurrence of records is quite close. The majority of the records, 37% of the records, fall within the optimal sleep duration. At the same time, up to 28% of the records fall outside the optimal duration, representing an interval between 6 and 7 h. According to the Centers for Disease Control and Prevention (CDC), 6 h of sleep is insufficient for most people, which defines short sleep as less than 7 h per night [43]. About 1/3 of adults sleep 6 h or less each night, and surveys suggest that short sleep may be increasingly common. A small percentage of people require sleep beyond the recommended hours [44] for their age group. Although rare, some people need less than 6 h of sleep per night. Similarly, others may need more than 9 h. Researchers believe genetics affects a person's ability to handle short sleep [45]. Quality sleep is essential for physical and mental well-being, immune system performance, hunger control, and cell and tissue repair. People may have sleep loss symptoms without understanding that they result from insufficient sleep. The respondent confirmed that he loses productivity during the day and often feels hungry. Quality sleep is essential for physical and mental well-being, immune system performance, hunger control, and cell and tissue repair. People may have sleep loss symptoms without understanding that they result from insufficient sleep. Recommendations on sleep duration are appropriate for guiding from a population perspective, whereas advice at an individual level (e.g., in the clinic) should be individualized to each person's realities.

The general assumption is that individuals get the right amount of sleep if they wake up feeling well-rested and perform well during the day. In addition to the amount of sleep, other important sleep characteristics such as sleep quality and sleep timing (sleep and wake times) should also be considered [46]. In conclusion, significant interindividual variability in sleep needs of sleep across the life cycle means that there is no "magic number" for ideal sleep duration. Based on feedback, not a single respondent suffered from poor sleep. Based on wearable device data and feedback from respondents, we can confirm the accuracy of wearable device measurements in sleep analysis.

### 4.2.2. Interruption of Sleep

When we think about sleep and health, we usually focus on the question of how much sleep we get and whether we are getting the recommended number of hours. Although total sleep time is undoubtedly important, sleep continuity, or avoiding interrupted sleep, is also crucial. The main symptom of intermittent sleep is easy for many people to observe: waking up from sleep 1 or more times at night. The implications of interrupted sleep can be significant and affect the quality of sleep and many aspects of an individual's health. In Table 7, we have shown the frequencies of how many times most commonly a user wakes up during sleep.

**Table 7.** Further results.

| ID | Interruption of Sleep | Average BPM Values | Time Interval |
|----|----|----|----|
| 11 | 1x | 40–50 | 05:00–05:29 |
| 23 | 3x | 45–55 | 06:00–06:29 |
| 25 | 6–10x | 30–40 | 03:30–03:59, 06:00–06:29 |

For respondent ID-11, 27% of the records represent a single interruption during sleep or 94 records out of a total of 349 records. For this respondent, we could determine the frequency of sleep interruptions from 0 to 3 times. Then, this would represent up to 72% of all records, or 251 records out of a total of 349 records. Respondent ID-23 has the most frequently recorded sleep interruption 3 times in one night, precisely 22% of all records, or 71 records out of a total of 327 records. This is immediately followed by 5 times (19%, or 63 records out of a total of 327 records) and one time (18%, or 58 records out of a total of 327 records) in one night, and when the three categories are combined again, in this case, it accounts for 59% of the total number of records or 192 records out of a total of 327 records. For respondent ID-25, almost all of the records were interruptions equal to 5 or more times (95%, or 493 records out of a total of 520 records) in a single night. By displaying a smaller interval for the previous two respondents, we were only able to see that, in this case, almost all of the entries are in the "5 or more times" category in the result. It was necessary to look more at those data, and in the result, we see that up to 58% of the records, or 300 records out of a total of 520 records are in the category "6 to 10 times" sleep interruption during one night. To be completely accurate, respondent ID-25 had the most interrupted sleep 8 times in one night, as can be seen in Table 8.

**Table 8.** Number of interruptions for respondent ID-25.

| Number of Interruptions | Abundance | Percentage Representation |
|----|----|----|
| 6 | 44 | 8% |
| 7 | 59 | 11% |
| 8 | 54 | 10% |
| 9 | 62 | 12% |
| 10 | 81 | 16% |

Importantly, these interruptions were predominantly in the morning, meaning that these results do not affect a given respondent's sleep quality. These results represent information regarding the respondent's morning wake-up time. Based on these results, we can say that there is a high probability that, for example, the respondent does not get up immediately at ringing the alarm clock.

4.2.3. Average BPM Values during Sleep

We plotted the average BPM values (recorded during sleep only) for every measurement. During sleep, it is normal for a person's heart rate to slow below the range of a typical resting heart rate. A normal heart rate can range from 40 to 100 beats per minute and is still considered average [47]. It can also vary from day to day, depending on hydration levels, altitude, physical activity, and body temperature. However, heart rate also changes during sleep as the sleeper goes through different stages of sleep. In the first stages of light sleep, the heart rate begins to slow down. During deep sleep, the heart rate reaches its lowest value. In REM sleep (sometimes referred to as paradoxical sleep), the heart rate may accelerate to a heart rate similar to that of wakefulness. By analyzing the average BPM values during each sleep stage, wearable devices and other monitoring tools can provide insights into the quality and duration of each sleep stage. In addition to sleep stage analysis, analyzing the average BPM values during sleep can also help identify potential sleep disorders and other health issues. For example, if the average BPM values are consistently elevated during sleep, this may be a sign of sleep apnea, a condition in which an individual's breathing repeatedly stops and starts during sleep. Sleep apnea can lead to a range of

health issues, including high blood pressure, heart disease, and stroke [48]. Similarly, if the BPM values are consistently low, it may indicate that the person is not getting enough physical activity during the day or may have an underlying health condition.

To better represent the results in the graph, we chose intervals of 5 values on the x-axis instead of accumulating a large number of records. Once the data we collected, the average BPM values we calculated over time, a night's sleep worth of sleep. In the graphs, we set boundaries because the average heart rate ranges from 60 to 100 BPM. However, during sleep, the average heart rate drops to 40 to 50 BPM, but again, this can vary [49]. Figure 4 shows that the average heart rate during sleep drops compared to the normal average heart rate.

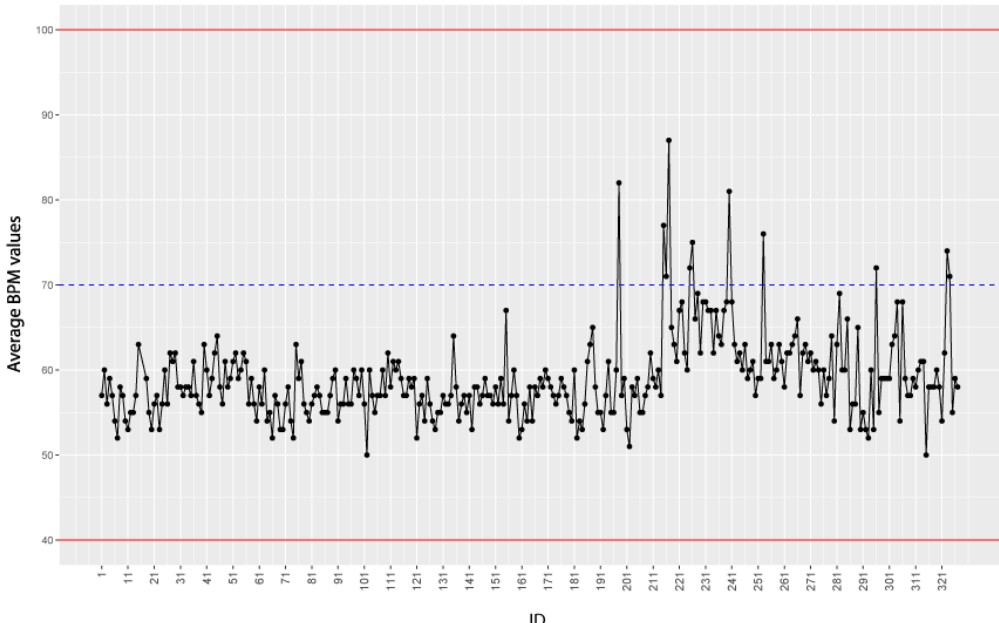

**Figure 4.** Average BPM values during sleep ID-23.

Figure 4 also shows a significant progression of average heart rate values during sleep. We explored possible causes where we considered potential factors contributing to the trend, such as changes in lifestyle, stress levels, medication use or medical conditions. Based on feedback from the respondent, we do not have data on the use of any medication. However, the respondent experienced a more challenging period when he started working during his studies, and his private life also changed. Based on his information, we took a closer look at the data, and in Table 9, we show the data of the top 10 values of average heart rate during sleep. We can see the dates on which those values were measured and have also matched the corresponding days based on the date. Of these data, 90% were measured on weekdays when the respondent also went to work during the study. Analyzing health data from wearable devices contributes to keeping users healthy. The presented output lets the user see how specific factors affect his health status in real-time. In addition, based on the results, we can alert users about changes in trends, which, if consulted with appropriate physicians, can help diagnose the user's health condition early. The results confirm the hypothesis we find in several articles and studies [42,50]:

**Hypothesis 3.** *Several factors can affect your sleeping heart rate, making it go higher or lower.*

Researchers presented survey results where up to 43% of adults report losing sleep quality or disrupting their healthy sleep patterns due to stress [51]. As a result of their lack of sleep, these people's stress levels grew, which can lead to a vicious cycle. Lack of sleep causes an increase in cortisol, the stress hormone, which causes stress to rise [52].

Adrenaline levels increase as a result of the body's normal stress response [53]. The interaction of these two hormones can increase blood pressure, blood sugar, and heart rate.

Overall, it is important to carefully analyze any significant progression of average heart rate data during sleep and take appropriate action based on the findings to ensure optimal health and well-being.

**Table 9.** Average BPM values during sleep (ID-23).

| Date | Day | Average BPM |
|---|---|---|
| 27 September 2021 | Monday | 87 |
| 18 August 2021 | Wednesday | 82 |
| 8 November 2021 | Monday | 81 |
| 21 September 2021 | Tuesday | 77 |
| 11 December 2021 | Sunday | 76 |
| 14 October 2021 | Thursday | 75 |
| 8 April 2022 | Friday | 74 |
| 13 October 2021 | Wednesday | 72 |
| 8 March 2022 | Tuesday | 72 |
| 24 September 2021 | Friday | 71 |

4.2.4. Min BPM Recording Time during Sleep

The last graph within this chapter presents us with the time interval and frequency of the minimum heart rate value. That is, when, in which period, a given respondent had a so-called "deep sleep". Deep sleep is the third stage of the sleep cycle associated with the slowest brain waves [54]. It is characterised by a slowing of the heart rate, breathing and complete relaxation of the muscles. It is during this phase that overall body regeneration, growth and detoxification occur [55]. An important piece of information is that waking up in this stage of sleep is difficult and uncomfortable for us. Therefore, it is a good idea to find out approximately when you are in this phase. Scientific studies show that:

**Hypothesis 4.** *Adults need less than 2 h of deep sleep [56].*

This means that 20% of our total sleep should be so-called deep sleep [57]. The need for deep sleep, however, is of course dependent on the age of the person and is just as sleep itself, an individual matter. We plotted the minimum heart rate values against the duration of sleep to visualize any patterns and trends. In Figure 5, we see the sinusoid from these data already shown, and we can tell that the most frequent occurrence of the minimum heart rate value is in the interval 06:00–06:29. The other two adjacent intervals contain a high number of occurrences of the minimum heart rate value of a given respondent. Our hypothesis was that every adult should be in the REM phase for some 2 h, and the wearable devices would confirm this by measuring the minimum heart rate value long enough. Thus, we can say that from approximately 05:00 to 07:29 the respondent is in the deep sleep phase, which covers the aforementioned 2 h.

Our analysis showed that all 3 respondents had minimum heart rate values during sleep that lasted long enough to confirm our hypothesis. The minimum heart rate values were consistent with the deep sleep phase, which typically lasts for 2 h per night. In addition, we observed that the minimum heart rate values were lowest during the first half of the night and gradually increased during the second half of the night. This pattern is consistent with the normal sleep architecture, where the first half of the night is dominated by deep sleep, while the second half of the night is dominated by REM sleep.

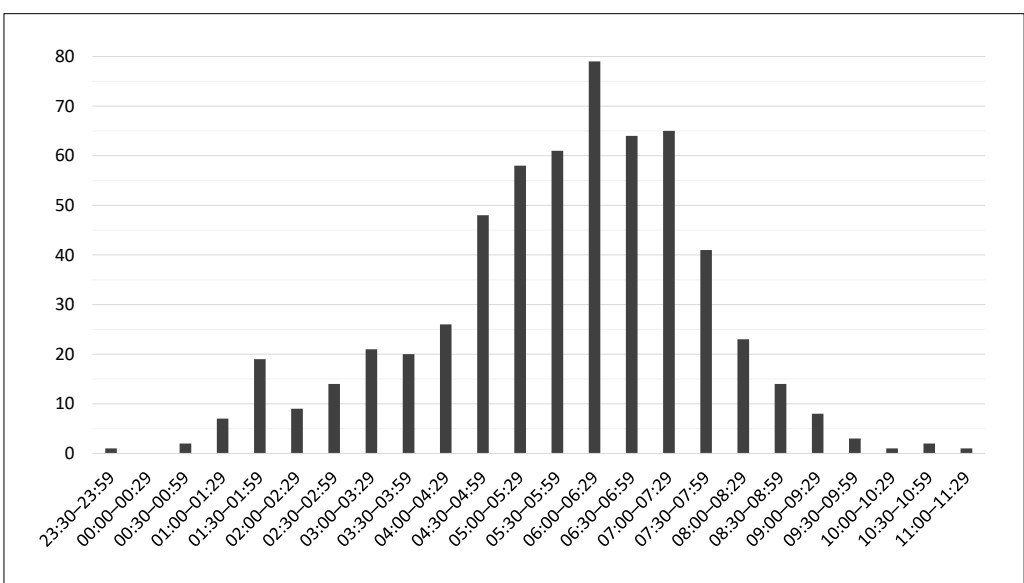

**Figure 5.** Min BPM recording time during sleep ID-23.

Our study demonstrates the importance of tracking minimum heart rate values during sleep to gain insights into sleep quality and overall health. Wearable devices equipped with heart rate sensors provide a non-invasive and convenient way to monitor heart rate values during sleep. These insights can be used to develop personalized interventions and treatment plans to improve an individual's cardiovascular health and overall well-being. By analyzing health data from wearable devices, we can generate personalized insights and recommendations for improving sleep quality. For example, we can identify specific factors that may be disrupting an individual's sleep, such as noise or light exposure, and provide recommendations for addressing these factors. Future research may reveal some knowledge about which outside variables negatively impact sleep or positively, helping to promote better and higher-quality sleep.

### 4.3. Workouts

We had two respondents of the same age and gender, so we decided to take a closer look at their results from the same types of workouts. We will compare two workouts: outdoor walking and functional strength training.

### 4.3.1. Functional Strength Training

In Table 10, we compared calories burned, but this is more for illustrative purposes only, as the ID-11 respondent has 77% more records than the ID-12 respondent. This information greatly influences the results of calories burned for us to be able to evaluate any analyses. As was the case with heart rate for the respondent with ID-12, we see larger values in all three columns, which also fits based on the larger heart rate values and based on weight information. For the values of maximum calories burned, we have a difference of 89.12 kcal, that is, there was a 9% greater value of maximum calories burned in any of the functional strength training sessions. The percentage difference in terms of the weight of the respondents is also 9%, which is indicative of the accuracy of the measurement of the wearable devices.

**Table 10.** Calories burned [kcal]—functional strength training.

| ID | Max | Mean | Median |
|----|-----|------|--------|
| 11 | 428.22 | 202.13 | 205.93 |
| 21 | 517.34 | 261.57 | 256.8 |

### 4.3.2. Outdoor Walking

Monitoring heart rate while walking can provide valuable insights into the cardiovascular fitness and overall health of an individual. By analyzing the data from wearable devices and comparing the readings of two respondents, you may be able to observe differences in heart rate patterns that could indicate variations in physical fitness levels, cardiovascular health, or other health conditions. In Table 11, we can see that within this type of training, the order has already changed between the respondents. The maximum heart rate was recorded in respondent ID-11, up to 191 BPM, and respondent ID-12 182 BPM. More important are the values of the average heart rate, which also help us to keep track of the results during training, and based on the study of [58,59], we can confirm that these values of the average heart rate in the age category of 20 years are within the norm, as they are in the interval from 100 BPM to 170 BPM. This implies that if the heart rate stays within this interval during further workouts of this type, there will be an improvement in the fitness of the respondents. Respondent ID-11 consistently has a higher heart rate than the other respondent ID-12 during the same walking exercise. This could indicate that he is not as physically fit as the other person or experiencing cardiovascular stress.

Based on the heart rate values, we expected higher values regarding calories burned in the respondent with ID-11. We observed more significant differences than we did within heart rate, with the most significant difference shown in median calories burned, where the difference is up to 90 kcal, which is 24%. We have a difference of 7% with average values of calories burned. These data are primarily influenced by the time the training is performed and the number of records.

**Table 11.** Heart rate [bpm]—outdoor walk.

| ID | Max | Mean | Median |
|----|-----|------|--------|
| 11 | 191 | 119.01 | 117 |
| 12 | 182 | 117.72 | 115 |

## 5. Conclusions

The limited number of respondents in this study should be considered in light of the pioneering nature of analyzing health data from wearable devices. The emerging field and potential technological constraints likely contributed to the smaller sample size. However, this study represents a significant step forward in exploring the feasibility and benefits of wearable devices for health monitoring.

The analysis focused on the heart rate, sleep patterns, and specific workouts of the respondents. Results indicated that heart rate values during functional strength training fell within the target zone, with variations observed between different types of workouts. Sleep patterns were found to be individualized, with variations in sleep interruptions among respondents. The study also highlighted the impact of individual factors, such as demographics and manually defined information, on workout outcomes.

Based on these findings, there is considerable potential to develop a cognitive healthcare platform that utilizes wearable devices. The platform can utilize the analyzed data from wearable devices to provide users with real-time health insights, detect anomalies, and send timely reports to individuals and their healthcare providers. This technology can empower individuals to proactively manage their health and receive personalized recommendations for improved well-being. In addition, other possible practical applications of the results revealed in this study could be:

- Personalized Health Monitoring: The analysis of heart rate, sleep patterns, and specific workouts provides valuable information for personalized health monitoring. The findings can be utilized to develop personalized health plans and interventions based on individual data, helping individuals make informed decisions about their fitness routines, sleep habits, and overall well-being.

- Fitness and Training Optimization: Understanding the variations in heart rate and workout outcomes between different types of exercises can aid in optimizing fitness and training regimens. Fitness professionals and trainers can utilize this information to tailor workout programs based on individual needs, maximizing the effectiveness and efficiency of exercise routines.
- Sleep Quality Assessment: The study's analysis of sleep patterns and interruptions contributes to assessing and improving sleep quality. By identifying factors that disrupt sleep, individuals can make necessary adjustments to their sleep environment or habits to enhance sleep quality, leading to better overall health and well-being.

These applications have the potential to enhance individual health outcomes and promote a more proactive approach to health management. In summary, despite the limited number of respondents, this study contributes valuable insights into the analysis of health data from wearable devices. The findings underscore the need for continued exploration in this emerging field and the potential for developing innovative healthcare solutions that empower individuals to monitor and manage their health effectively.

**Author Contributions:** Conceptualization and methodology, N.H. and E.K.; formal analysis and supervision, E.K. and I.Z.; writing—original draft preparation, N.H.; writing—review and editing, N.H., E.K. and I.Z.; funding acquisition and project administration I.Z. All authors have read and agreed to the published version of the manuscript.

**Funding:** This work was supported by the Slovak Scientific Grant Agency VEGA under the contract 2/0135/23 (30%) Intelligent sensor systems and data processing and by EDEN: EDge-Enabled intelligeNt systems, project number VEGA 1/0480/22 (70%).

**Data Availability Statement:** The data presented in this study are available on request from the corresponding author. The data are not publicly available due to the nature of the data (private health data of respondents).

**Conflicts of Interest:** The authors declare no conflict of interest. The funders had no role in the design of the study; in the collection, analyses, or interpretation of data; in the writing of the manuscript; or in the decision to publish the results.

## Abbreviations

The following abbreviations are used in this manuscript:

| | |
|---|---|
| AHA | American Heart Association |
| AI | Artificial Intelligence |
| BPM | Beats Per Minute |
| CDC | Centers for Disease Control and Prevention |
| COVID19 | Coronavirus Pandemic |
| CRISP-DM | CRoss-Industry Standard Process for Data Mining |
| ECG | Electrocardiography |
| ETL | Extract, Transform, and Load |
| GDPR | General Data Protection Regulation |
| HR | Heart Rate |
| HRV | Heart Rate Variability |
| IoMT | Internet of Medical Things |
| IOT | Internet of Things |
| LoA | Limits of Agreement |
| MHR | Maximum Heart Rate |
| PPG | Photoplethysmography |
| PRV | Pulse Rate Variability |
| REM | Rapid Eye Movement Sleep |
| RPE | Rating of Perceived Exertion |
| WHO | World Health Organization |

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
