# Peer review of "A Validation Study to Confirm the Accuracy of Wearable Devices Based on Health Data Analysis"

_electronics, doi:10.3390/electronics12112536_

Round 1
Reviewer 1 Report
Major revision is required for this manuscript before consideration for publication.
1. The data organization in the manuscript should be improved. For example, the meaning of recorded data in Table 1 and 4 are unclear; In line 367,"Although subjective, rating exertion on a rating scale of 6 to 20, as seen in Table 3”,it should be Figure 3; Figure 5 is not discussed in the manuscript.
2. In Line 343, 'We have distorted the data and plotted it from the date 19/11/2021 to 20/06/2021 to make the results more visible' while it was not included in Figure 2.
3.In Line 361, 'At Figure 4, we can see a comparison of 2 physical activities based
on their maximum heart rate' while Figure 4 shows average BPM values.
4. The supporting data for those values of "27%, 72%, 22% ''etc in Section 4.2.2 are missing.
5. In Figure 9, BMP values are recorded while it was illustrated as Calories burned.
should be polished.
Reviewer 2 Report
please see the attachment

Round 2
Reviewer 1 Report
A point to point response letter is normally required
Ordinarily written.
Reviewer 2 Report
All major comments were adequately addressed and the Authors have done an admirable job of improving the quality of the manuscript. Therefore, it can be accepted without any structural modification.